# Critical influence of implicit motion features on hand-drawn lines' emotional expression

Xinya Liu*
School of Artificial Intelligence
and Computer Science,
Jiangnan University,
and Jiangsu Key Laboratory
of Media Design and
Software Technology

Ruimin Lyu†
School of Artificial Intelligence
and Computer Science,
Jiangnan University,
and Jiangsu Key Laboratory
of Media Design and
Software Technology

Huan Liu‡
School of Artificial Intelligence
and Computer Science,
Jiangnan University,
and Jiangsu Key Laboratory
of Media Design and
Software Technology

Guoying Yang§
Goldsmiths College,
University of London

Yuefeng Ze¶
New Media College ,
Shanghai Institute of Visual Arts

## ABSTRACT

As a basic visual element, the way lines express and evoke human emotional responses is an important topic of interest in fields such as cognitive psychology and human-computer interaction. In the past, many studies have attempted to identify the relationship between the static features of lines and emotions but have overlooked the motion features of hand-drawn lines. However, some recent studies have shown that the motion behaviour implied by hand-drawn lines has a significant impact on viewers' aesthetic-emotional judgments. In this study, we sought to explore the motion features that have a key impact when people actively express emotions through drawing lines. In the experiment, subjects actively expressed specified emotions by drawing lines by hand, and their pen movement data were fully recorded on a digital screen. Several static and motion features were estimated using the experimental data. Moreover, we quantified their correlations with emotional pleasure and arousal through multiple ordered logistic regression analysis. The resulting static and motion features that significantly affect the emotional expression of lines were identified. In addition, a variety of motion features were crucial in the emotional expression of hand-drawn lines. This could help guide artists and designers in their creative work and has the potential to be used in the development of emotional computing applications for line, picture, interactive art, and graphic design.

**Index Terms:** Human-centered computing—Human computer interaction (HCI)—Empirical studies in HCI; Applied computing—Arts and humanities—Fine arts

## 1 INTRODUCTION

In recent years, with the upgrading of intelligent interactive surface and the increasing demand for emotional care in human-computer interaction, psychological research in human-computer interaction has been further developed. People are trying to investigate the traditional proposition of the influence of very basic visual elements on human aesthetics and emotions by using more advanced equipment and experimental methods [4, 19, 30, 38].

*e-mail: xinya.liu@stu.jiangnan.edu.cn
†e-mail: ruiminlyu@jiangnan.edu.cn
‡e-mail: 6213113012@stu.jiangnan.edu.cn
§e-mail: G.Yang@gold.ac.uk
¶e-mail: zeyuefeng118@qq.com

Line is an essential carrier of visual interaction and a basic component of many visual arts and graphic design in the East and the West. And it frequently appears as outlines in the process of visual interaction between human beings and external things. How lines express and arouse human emotions has always been the focus of attention in the fields of human-computer interaction, visual communication, and cognitive psychology. The cognitive emotion of lines has been the subject of empirical research by numerous researchers. These works mainly focus on finding the relationship between static geometric features such as angle, thickness, direction, and straightness of lines and human emotions [1, 18, 33, 40, 41]. In our daily experience, when we encounter artworks, visual design and interactive graphics, it makes us feel more human if we find that hand-drawn lines are used in these works. This emotional response cannot be explained simply by the static features of the lines.

We speculate that the "motion" factor in hand-drawn lines is likely to be the key to this "human touch" feeling. In fact, this speculation can be found in many previous visual artworks and literature reports from both East and West. The art of calligraphy and painting in the East particularly emphasizes and pays attention to "brushwork", that is, the method of using the brush. Its core is to write a specific form of "brush strokes" by using certain brush skills, so that it can convey rich motion sensation. In the West, since Impressionism, many artists have consciously or unconsciously trained unique brushwork skills. Degas, Van Gogh, Pollock, Willem de Kooning, Franz Klein, etc. are a few examples. Even so, it can be argued that these implicit brushwork movements in their works have achieved their unique styles. Although the "brush strokes" in the completed works are all presented as static images, what is conveyed to the viewer through these works contains rich feelings of movement. In light of the foregoing, the following query is raised by this article: **In artworks and visual interactions, can those implicit motion features play a key role in the emotional expression of hand-drawn lines? What connection do they have to emotions?**

For the above questions, there is still a lack of directly relevant empirical research. However, in some adjacent fields we found some enlightening results. In the aesthetic models proposed in recent years, many works listed the motion sensation of handwritten lines as an important factor [13, 23, 49, 53], and some experimental studies also confirmed that motion perception plays an important role in aesthetic judgment [6, 22, 34, 44, 48, 50]. And as early as 1921 in a systematic study on the emotional perception of hand-drawn lines, Lundholm [33] proposed that the drawing process of lines may be related to the emotional expression. He believed that recording motion data such as speed and pressure during the painting process is a

direction that needs attention in follow-up research. People have recently been able to use cutting-edge equipment to capture the motion data of hand-drawn lines thanks to the continuous development of interactive technology. However, the study of the relationship between the motion features and the emotional expression of the lines has remained almost undeveloped. Scholars in the field of handwriting affective computing are also aware of the rich emotional information contained in the motion data during writing and drawing. And they have conducted a great deal of research using modern devices to capture human handwriting movements and attempt to identify hidden emotions from them [2, 11, 12, 15, 28, 29, 37]. However, such studies are biased towards passive emotion induction rather than active ones, and the recognised emotions are implicit. And in almost all artistic endeavours, the creator uses active emotional expression. The results of research in the field of handwriting affective computing are therefore difficult to transfer to the study of emotion mapping in line or other artworks. Taken together, these works on the one hand indirectly affirm our hypothesis that the motion features of hand-drawn lines may be deeply related to emotional expression. On the other hand, they point to a direction in which we can improve: we can use modern equipment to capture data on people's movements when actively expressing emotions through drawing lines, to uncover key features in them, and to ascertain how they relate to emotions. Our experiments and contributions are described below.

Firstly, we designed a smart hand-drawn line emotional expression experiment. During the experiment, the subjects can actively express their emotions on the digital screen by hand-drawing lines, while the digital screen simultaneously records the pen movement data. . In this way, we have established an online hand-drawn line dataset. The experiment's coverage of emotional data is, as far as we are aware, the most comprehensive. Subsequently, we extracted extremely rich features from the original data, and established a model from features to emotional responses through multiple ordered logistic regression analysis. More critically, the model has screened out the significant features that affect the emotional expression of hand-drawn lines. Some of the static features are consistent with previous studies, thus verifying the validity of the experimental data. But more importantly, we found some very important motion features that were previously overlooked and that have a significant influence on emotional expression. Our study has implications for guiding artists and designers in their creative work and has potential applications for developing affective computing applications regarding line, image, interactive art, and graphic design. It is important to emphasise that the research in this paper focuses on the motion influencing factors on the expression of emotion through the line from the creator's point of view, and does not yet address the motion influencing factors that affect the viewer's perception. That might be part of our future work but cannot be mentioned in this report for the time being.

## 2 RELATED WORK

### 2.1 The Emotional Metaphor of the Static Features of the Lines and Their Derived Forms

There have been many empirical studies on the mapping relationship between the static geometric features of lines and their emotional expression. The phenomenon that different styles of lines can express different emotions was first empirically investigated by Lundholm [33] in 1921. He selected 48 adjectives and asked the subjects to express them using hand-drawn lines. He grouped the adjectives into 13 sets and found that lines corresponding to synonyms of the same set showed significant similarity among subjects. Sharp angles were typically associated with negative and powerful emotions, whereas rounded angles were typically associated with positive and tranquil emotions.

A series of empirical studies have since focused on comparing curved, straight, and broken lines for emotional expression. Hevner's study [18] correlated straight lines and curves with different emotional words. Aronoff et al.'s study [1] showed that curves were more aesthetically pleasing than straight or angled lines. Human emotional preferences for curves have been confirmed in various later research [3, 5, 7, 38], even in cross-cultural surveys [14] and investigations of great apes [36]. Bertamini et al. [4] further found that people simply showed a greater preference for curves without an aversion to angles. And the effect of people's preference for curves may have influenced the emotional expression of some visual media [24, 43, 47, 56].

It has also been found that static features other than the curvature features of lines also affect emotional expression, such as the angle, direction, and spatial distribution of lines. Direction upwards represents strength, energy, force, ambition, uplifting feelings, tiredness, pressure, etc., while direction downwards represents weakness, lack of energy, relaxation, depression, etc. [40]. Heavy lines are full of strength, and light lines are weak [41]. The number and clutter of lines affect harmony and pleasure [45]. These findings on the mapping relationship between geometric features of lines and their emotional expressions have guided both artistic creation and design.

The above studies also inspired affective computing on images based on line features. Based on the findings of the correlation between line orientation and the feelings of stability and motion, Wang et al. [55] proposed a feature vector named WLDLV to classify the emotional semantics of images with good results. Based on the findings that the complexity and curvature of lines affect emotional expression, Lu et al. [32] extracted contours from complex images and calculated relevant features such as roundness-angle and simplicity-complexity to enhance the emotional classification of images.

### 2.2 Empirical studies on the subjective motion of hand-drawn lines in aesthetic appreciation

Aesthetics is a high-level human intelligence, and modelling it is a rather challenging task. Among the various proposed models of aesthetic appreciation, some works ignored "motion" [25, 31, 42], but more works included it as an important factor in the model [13, 23, 53].

In experimental studies on the motion perception of hand-drawn lines, some have demonstrated that hand-drawn lines do significantly trigger "motion imitation" neural activity in the viewer's brain and influence aesthetic judgments. Knoblich et al. [22] found that the observation of static graphic symbols evokes the motion simulation of the gestures needed to produce the symbols. Through EEG, Umiltà et al. [50] and Sbriscia-Fioretti et al. [44] showed that the brain's motor system is actively involved in aesthetic behaviour when viewing Lucio Fontana's and Franz Kline's abstract paintings composed of unique brush strokes. Ticini et al. [48] found that consciously imitating the artist's actions when viewing artworks can enhance the aesthetic experience. Lyu et al. [34] proposed a method for measuring the subjective movement of brush strokes, illustrating that the motion features of hand-drawn lines significantly influence the viewer's judgement of artistic style. Rebecca et al. showed experimentally that people prefer lines that reflect the natural movement of the human body and that the preference is positively correlated with the level of artistic training [6].

These studies undoubtedly suggest that the motion features of hand-drawn lines may be closely related to human cognition and emotional expression.

### 2.3 Preliminary Studies on the Emotional Metaphor of the Motion Features of Lines

Previous studies have mostly ignored the motion features of hand-drawn lines, but some researchers have realised the hidden relationship between the implied motion of lines and their emotional expressions [19, 20, 52].

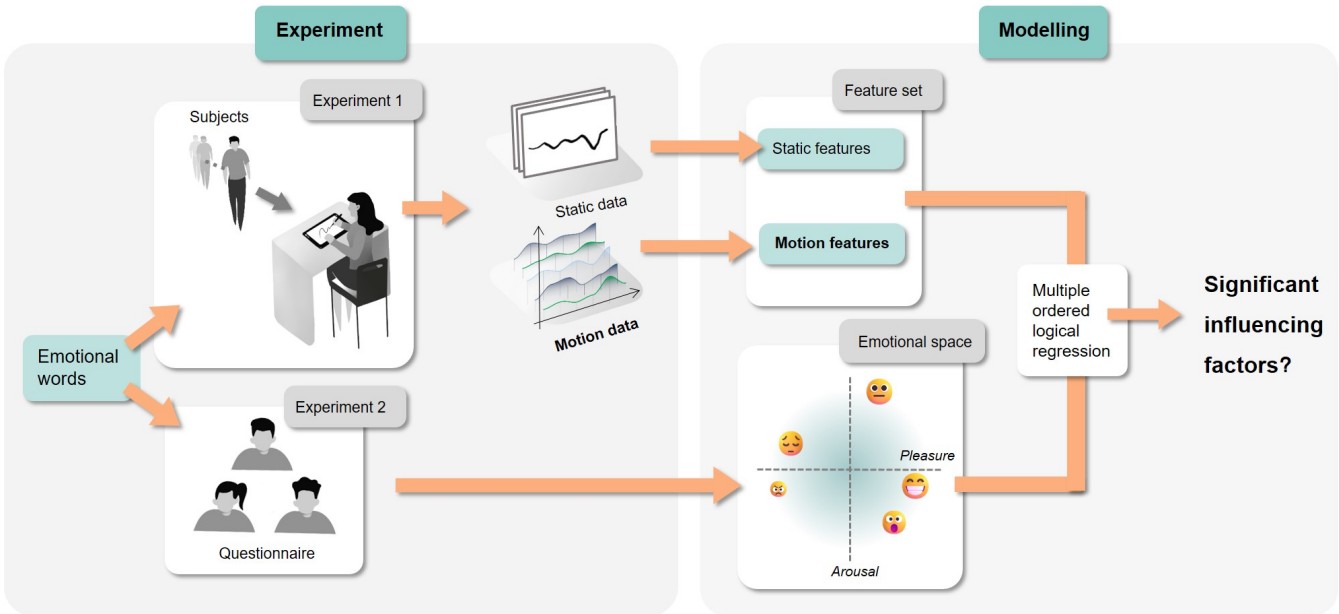

Figure 1: Research framework

Lundholm [33] observed that lines appeared to resemble the motion expression of emotional states. He speculated that there might be a close connexion between the process of drawing lines and the emotional states being expressed. He suggested that recording the speed and pressure during the drawing process is a direction to focus on in subsequent studies. Hu et al. [19] empirically investigated Lundholm's conjecture [33] and found that the procedurally generated static lines and their corresponding motion animations are highly correlated in terms of emotional expression, and there is a translational relationship in Russell's 2D emotion space. Ibáñez et al. [20] constructed a line-based model of emotional expression, in which the line's speed of movement is an important attribute influencing arousal.

It can be seen that, in addition to static geometric features, the motion features of the line formation process are also crucial to its emotional expression. With the advent of various digital interactive surfaces and motion-sensing devices, it has become possible to capture motion information during line drawing, but related research is still very scarce.

## 2.4 Handwriting Affective Computing

In the field of affective computing, some researchers have realised that motion data during writing can be used to identify latent emotions [2, 11, 12, 15, 28, 29, 37]. Such studies typically employ digital interactive devices to collect handwritten data from subjects, extract static and motion features, and build a model for emotion recognition. Likforman-Sulem et al. [28] built the first database for identifying anxious emotions from handwriting and drawings, and used random forests for feature selection and to identify emotions. They found that most of the selected features were correlated with online data, demonstrating the importance of motion data in identifying emotions. Later, Nolazco-Flores et al. [37] extracted richer time-frequency domain features based on this dataset and trained a radial basis SVM model using the LOO method, resulting in a 15% improvement in the average classification accuracy compared to the baseline. Han et al. [15] recorded emotional handwriting data and extracted richer handwriting motion features including pressure, altitude, azimuth, etc. Thus, they extended handwriting emotional recognition to the four quadrants of Russell's 2D emotion space.

The above studies were all intended to identify the underlying emotions in handwriting and therefore adopted a passive approach to emotion elicitation in data collection. However, in activities such as artistic creation, visual design, and interaction design, emotional expression is often intentional and performative [21, 54]. Passive emotion elicitation is not sufficiently broad and precise to cover various emotions. Therefore, it is necessary to explore the use of active emotion elicitation to construct a database of emotional hand-drawn lines.

## 3 RESEARCH FRAMEWORK

The research framework of this study is shown in Figure 1. The first step is data collection, which includes two experiments. Experiment One is a hand-drawn lines data collection experiment. Based on the research by Cowen and Keltner [8], we pre-selected 28 emotional words to cover the emotion space. The self-designed programme would randomly select emotional words for the subjects. The subjects then expressed their emotions on the digital screen by drawing lines according to the emotional words we provided. The backstage device would record the coordinates, pressure, altitude, and azimuth. At the same time, the corresponding static image data were generated according to the motion time series data. Experiment Two is a conversion experiment from discrete emotional words to dimensional emotion space. We intended to convert the emotional words used in Experiment One to Russell's 2D emotion space. A questionnaire was issued to the subjects to evaluate the dissimilarity among emotional words.

The second step is to model the relationship between the hand-drawn line features and the pleasure and arousal dimensions of Russell's 2D emotion space [24], and to extract the important features for discussion. From the questionnaire results in Experiment Two, we used the MDS (Multidimensional Scaling) algorithm to transform emotional words into Russell's 2D emotion space based on the dissimilarity among them. We then used a Gaussian mixture model to cluster the 28 emotional words into seven classes on the pleasure and arousal dimensions, respectively. From the original data in Experiment One, we extracted a wide range of static and motion features to form the feature set. Importantly, we also extracted a large number of frequency domain features. Compared to

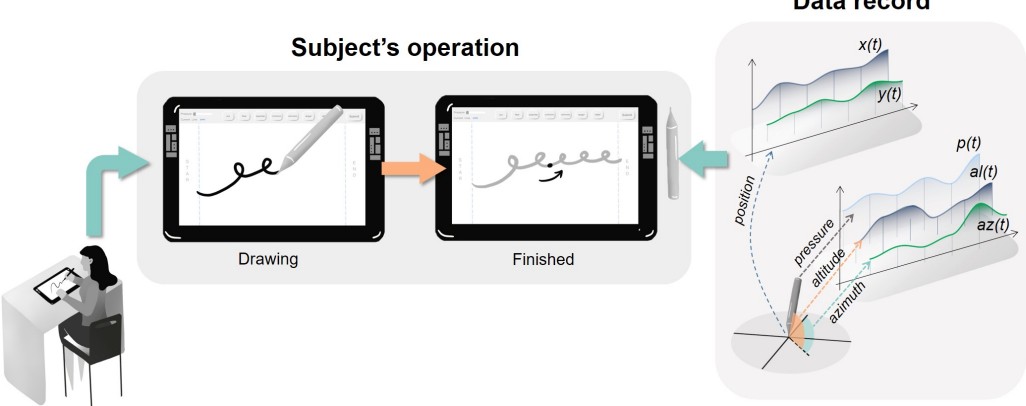

Figure 2: Operation and data collection of subjects in the collection experiment

previous studies, we covered more features. We then used distance correlation and random forest algorithms to reduce the feature set's dimensionality. Finally, we modelled the feature set with the pleasure and arousal dimensions separately through multiple ordered logistic regression. We found that there are critical features that have significant effects on pleasure and arousal, and there is a linear relationship between these features and the two dimensions.

## 4 EXPERIMENTS

### 4.1 Experiment One: Hand-drawn Emotional Lines Data Collection

The purpose of this experiment was to collect static and motion data when subjects used hand-drawn lines to express specific emotions. Considering the difficulty and deviation of using dimensional emotion models as labels for data collection, we designed a hand-drawn line emotion expression experiment based on discrete emotional words, similar to Lundholm's study [33]. The subjects were required to draw lines on a digital screen to express specific emotions (with emotional words shown on the same screen) using a data collection programme we developed. The programme captured and recorded associated motion data while the subjects drew the lines. We used the motion data to generate the static image data of the lines. We combined these static and motion data to form an emotional line dataset for feature extraction and modelling. The following section describes in detail the preparation of the experiment, the process of the experiment, and the processing of the original data.

#### 4.1.1 Experiment Preparation

**Hardware:** The surface material of the Wacom LCD digital screen is similar to the texture of paper, providing a more realistic hand-drawing experience than a tablet device. Additionally, the interaction between the digital screen and the subject is more similar to hand drawing than that of a digital board. Therefore, we used a Wacom DTZ-1200W LCD digital screen and a matching grip pen for the subject's line drawing. The screen has a size of 12.1 inches, a ratio of 16:10, and a resolution of 1280*800 pixels. The grip pen is sensitive to the tilt and pressure applied by the subject.

**Data Acquisition Programme:** The data collection programme was developed based on the Unity engine. The subject can use the grip pen in the drawing area to draw according to the displayed emotional word, which is shown on top of the interface. Our study focuses on the motion feature of lines. Therefore, the programme not only renders the static effects of the drawn lines but also uses a small ball to reproduce the motion of the pen tip at the end of each drawing for the subject's reference, as shown in Figure 2. The subjects are

allowed to draw many times until they are satisfied. As the object of our study is the hand-drawn line, we tried to avoid the subjects drawing other non-line styles as much as possible. To achieve this, we use one uninterrupted drawing session as one recording unit. The starting and ending points of the drawing need to be within the limits for a successful pass. The subjects can only submit when all the emotional words are drawn, and all the results are valid. The programme collects motion data with a fixed interval of 1/60s, including the x-coordinate, y-coordinate, pressure, altitude, and azimuth. The bottom left corner of the screen is used as the origin of the axes, with increasing x in the horizontal direction and increasing y in the vertical direction. Figure 2 shows an example of the collected motion data.

**Choice of Emotional Words:** We used 28 discrete emotional words as emotional labels for the subjects' samples, which almost covered all human emotions. Among them, 27 adjectives were taken from the study of Cowen and Keltner [3]: sadness, adoration, anger, awkwardness, nostalgia, anxiety, surprise, awe, craving, fear, joy, confusion, romance, satisfaction, calmness, admiration, relief, entrancement, horror, empathic pain, boredom, appreciation, excitement, sexual desire, disgust, interest, and amusement. An additional emotional arousal word "activation" was added to them.

#### 4.1.2 Collection Process

The main part of the data collection experiment was conducted during a classroom session. Considering the time and energy of the subjects, each subject was asked to draw seven emotional words. A total of 98 people participated in this experiment. Subsequently, to further expand the dataset, 18 volunteers were recruited to perform the same process for all 28 emotions. In summary, a total of 116 adults participated in the experiment (mean age: 21.4, age SD: 1.4; female: 56, male: 60; right-handed: 107, left-handed: 9).

Before the start of the experiment, the researcher gave all subjects a detailed introduction of the process. The subjects were also given the chance to ask questions to ensure they understood the purpose and process of the experiment. Then, the researcher informed the subjects of the emotional words they would need to express by hand-drawing in the formal experiment. The researcher provided the subjects with pencils and paper. The subjects used pencils to carry out an unrestrained rehearsal of expressing emotion through hand-drawn lines. This ensured that they were familiar with expressing emotions through hand-drawn lines and could draw more satisfactory effects in formal experiments. When the subjects felt they had no more questions and were fully prepared, they informed the researcher and signed ethics agreements. The researcher then instructed the subjects to begin the formal experiment.

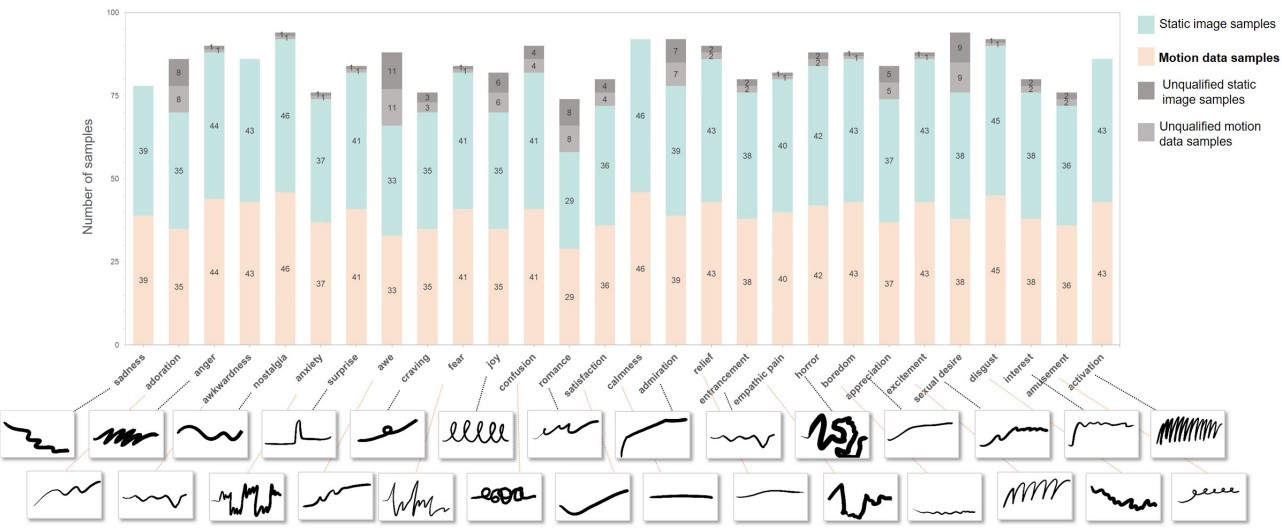

Figure 3: Distribution of original data set and images of typical samples

The formal procedure is shown in Figure 2. A subject was directed into a quiet room where the digital screen and grip pen were placed on the table. The acquisition program was turned on, and the subject was seated comfortably in front of the acquisition device. The subject then entered his or her personal information into the data collection programme and entered the experiment interface. The subject could select the prior emotional word he or she wished to draw from the menu bar at the top. He or she then used the grip pen to draw in the drawing area while the drawing data was recorded in the backstage. After each drawing, the program automatically displayed an animation of the drawing process for the subject's reference. The subject could draw an unlimited number of times until he or she was satisfied. After drawing the remaining emotional words, the subject clicked the submit button to finish the experiment.

It is particularly important to note that this experiment did not use video or other forms to elicit emotion from the subjects. As our study is not concerned with the potential emotional state of the subjects at the moment but with their intention to express their emotions through lines. Therefore, the subjects were instructed to express emotions through lines, rather than just drawing arbitrary lines in a certain mood.

### 4.1.3 Original Data Processing

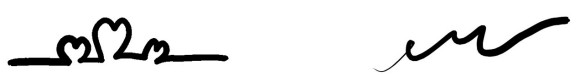

Express romance through symbol          Express romance through pure line

Figure 4: Unqualified line sample containing symbols and qualified line sample

**Data Filtering:** A total of 1190 original motion time series data were collected for the experiment, and static image data in JPG format were generated from the original motion time series data. During the collection process, we found that although the subjects were instructed to express the specified emotions using pure lines, some of them still used specific and common symbols [39] to express the emotional words. A typical example is shown in Figure 4, where the left-hand sample uses the symbol of love to express

romance, while the right-hand sample uses only a pure line. We removed samples that used symbols for each emotional word. Thus, 84 unqualified original motion time series data and corresponding static image data were eliminated. The distribution of unqualified samples is shown in Figure 3.

**Data Set Collation:** After completing the data filtering, we combined the static images of the qualified lines with the motion time series data into a complete original dataset. The distribution of the original dataset and typical samples within it are displayed in Figure 3. It is important to note in particular that given that our study was curious about how people express emotions through hand-drawn lines. We wanted to discover commonalities in how humans express emotions through hand-drawn lines. Therefore, data labels were given at the time of subjects collection and no additional subjects were recruited for a second round of attaching labels. Because that would involve the efficiency of hand-drawn lines conveying emotions between humans, rather than our original intention of focusing on the expresser. In the end, we obtained a total of 2212 qualified data, including 1106 static data in 1280*80 pixels JPG format and 1106 motion data in CSV format with a 1/60th of a second interval recording x-coordinate, y-coordinate, pressure, altitude, and azimuth.

### 4.2 Experiment Two: Transformation Experiment from Discrete Emotional Words to Dimensional Emotion Spaces

Russell's 2D emotion space is an accepted and widely used classical dimensional model in emotional research. Associating our study with this model can enhance the transferability, extensibility, and interpretability of our study. The aim of this experiment is, therefore, to convert the discrete emotional labels into dimensional emotion space coordinates. We collected dissimilarities among emotional words, which can be used to calculate the coordinates of our chosen emotional words in the 2D emotion space.

The experiment employed anonymous questionnaires to capture the dissimilarities among emotional words. The questionnaire contained 10 multiple-choice questions, with each randomly selecting one of the 28 emotional words. The subjects were asked to select 1-10 antonyms from the remaining 28 emotional adjectives. A total of 79 adults participated in this experiment (mean age: 21.8, age SD: 1.6; female: 40, male: 39).

## 5 MODELLING

This study explores the significant features that influence line arousal and pleasure, as well as the relationship between them. Therefore, regression models should be established separately between the features of lines and arousal and pleasure. However, in the transformation experiment of discrete emotional words to dimensional emotional space, we were able to obtain emotional coordinates corresponding to each type of emotional word rather than the emotional coordinates of each sample. Thus, we can use multiple ordered logistic regression, which is applicable to the case of ordered categorical labels for modelling. The modelling details are presented in this section.

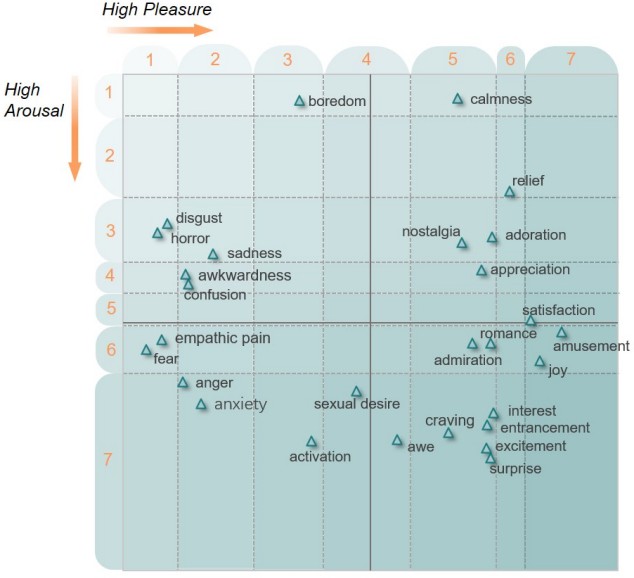

Figure 5: Emotion space and grades of clustering

### 5.1 Creating Emotion Space and Clustering Grades Using Gaussian Mixture Model

The greater the frequency of selection between two emotional words in the questionnaire of Experiment Two, the higher the dissimilarity. Based on the questionnaire results, we constructed a dissimilarity matrix between emotional words by calculating the frequency of pairwise selections. We then used the MDS algorithm, combined with the dissimilarity matrix, to create a distribution of the 28 emotional words in two dimensions, as shown in Figure 5.

We observed that emotional words related to high pleasure clustered around the positive edges of the x-axis, while those related to low pleasure clustered around the negative edges. On the y-axis, emotional words associated with low arousal clustered around the positive edges, while those associated with high arousal clustered around the negative edges. This model exhibits a high similarity to Russell's 2D emotion model, where the x-axis represents the pleasure dimension, and the y-axis represents the arousal dimension. Thus, we have successfully transformed discrete emotion words into a 2D emotion space.

Generally, the number of categories used in multiple ordered logistic regression should not exceed eight. Additionally, the use of multiple ordered logistic regression requires that the parallel line test is satisfied, which tests whether the effect of each value level of the independent variable on the dependent variable is the same across regression equations. We observed that the 28 emotional words were not evenly distributed on the x-axis and y-axis. Therefore, we

used the Gaussian mixture model to cluster the coordinates of the 28 emotional words on the x-axis (pleasure dimension) and y-axis (arousal dimension) into seven evenly increasing levels, respectively. The clustering results are shown in Figure 5.

### 5.2 Feature Extraction

To identify the most significant features associated with the emotional metaphor of lines, we need to consider as many potentially relevant features as possible. In addition to incorporating a priori image features [10, 18, 26, 33, 40, 51, 57] associated with line emotion mapping, we also included many motion features [2, 9–12, 15, 17, 26, 27, 35, 57] from previous studies on handwriting emotion computing. Furthermore, we explored additional potential features. Specifically, we took the first to fourth derivatives of the obtained original motion data to capture their features in terms of velocity, acceleration, jerk, spasm, etc. Additionally, we transformed the motion signals into the frequency domain to obtain a large number of statistical frequency domain features.

As described below, for each sample, the available original data include a binary image of the line, the horizontal position of the pen tip $x(t)$, the vertical position of the pen tip $y(t)$, the pressure of the pen $p(t)$, the azimuth $az(t)$, and the altitude $al(t)$. For the static image data, we calculated 16 classes of image features, partly based on a priori experience with the emotional metaphor of the line and partly from commonly used image processing features. For the motion time series data, we took the first to fourth derivatives of $x(t)$, $y(t)$, $p(t)$, $az(t)$, and $al(t)$ with respect to t to obtain their corresponding velocity $v(t)$, acceleration $a(t)$, jerk $j(t)$, and spasm $s(t)$. Based on each time domain signal, we calculated 29 types of general statistical features, as well as four independent time domain features, two of which are from hand-written features studies [10] not covered in the general statistics. We further converted the time domain signals into frequency domain signals and calculated the amplitude spectrum, energy spectrum, power spectrum, and power density spectrum of each time domain signal using the periodogram method, and the power density spectrum using the parameter estimation method. For each frequency domain signal, we calculated 21 general statistical features and five independent features for each of the two power density spectra. Finally, we calculated three entropy features for each time domain signal.

In summary, we extracted 3695 features for each sample. The feature extraction framework is shown in Figure 6. In the supplementary materials, we provide tables 5, 6, 7, and 8 to supplement details about the features. Table 5 describes all the raw data used for feature computation, table 6 describes the static features we extracted, and tables 7 and 8 describe the motion features we extracted. In tables 6, 7, and 8, we describe where each feature is extracted from, and in conjunction with table 5, readers can understand how these features were obtained. Additionally, we also note in tables 6, 7, and 8 which features come from previous research.

### 5.3 Feature Dimensionality Reduction Using Distance Correlation and Random Forests

The need to explore the relationship between the emotion space and a large set of features necessarily requires dimensionality reduction, which involves eliminating useless features and merging related features. We started by filling in the gaps and removing features with zero variance of the independent variable. The number of features was reduced to 3691.

Some of the features we extracted were linearly related to each other. The linearity of the independent variables could lead to multiple collinearity problems in modelling. This could lead to poor model fit and also make it difficult to interpret the model. Therefore, we first calculated the distance correlation [46] between each pair of features. The distance correlation coefficient ranges between 0 and 1, with a stronger correlation closer to 1. To merge as many

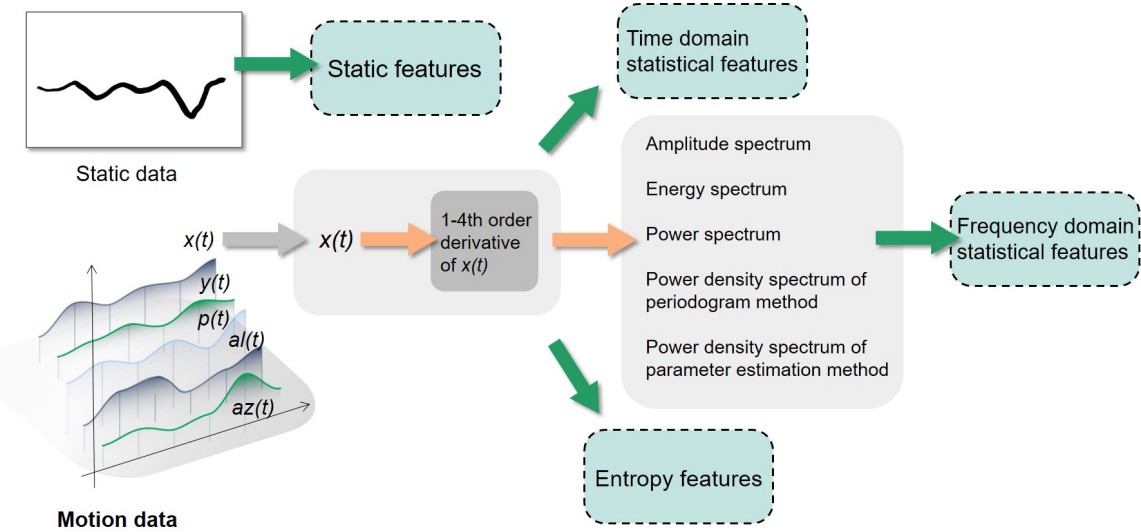

Figure 6: Feature extraction framework

redundant features as possible while avoiding the erroneous merging of irrelevant features, we chose a moderate threshold of 0.6 to merge features. As a result, the size of the feature set was successfully reduced from 3691 to 326.

However, for the current sample size, the number of independent variables is still high for modelling. Based on the study by Harrell et al. [16], regression models built by keeping the EPV (events per variable) between 15 and 20 would be safer and more reliable. To include the right number of independent variables and to involve the most relevant features in modelling, we used a random forest algorithm to rank the importance of features in relation to the ordered categorical emotional labels. We then filtered out 66 features that were most relevant to the pleasure dimension and 60 features that were most relevant to the arousal dimension.

## 5.4 Establishing Linear Models between Features and Arousal and Pleasure

We used the filtered feature values of the 1106 samples as independent variables and the seven ordered classes of arousal and pleasure obtained by clustering as dependent variables to establish multiple ordered logistic regression models. The main goal of building these models was not to predict, but to discover the significant factors influencing line emotion. Thus, we focused our analysis on the relationship between significant features and the influence of emotional dimensions.

## 6 Result Analysis and Discussion

### 6.1 Model Fitting and Significant Features

The established pleasure dimension model and arousal dimension model both passed the likelihood ratio test to determine whether the partial regression coefficients of whether the partial regression coefficients of all independent variables were all 0 ($p < .05$), as well as the goodness-of-fit test (Pearson $p > .05$, Deviance $p > .05$), and satisfied the parallel line test ($p > .05$). Collectively, the two multiple ordered logistic regression models established in this study passed the tests and had strong explanatory significance in the regression results.

The information of the ordered logistic regression model built with the feature set obtained from random forest screening as independent variables and the seven-level pleasure dimension labels

as dependent variables are presented in Table 1. Similarly, the information of the ordered logistic regression model built with the seven-level arousal dimension labels as the dependent variable is presented in Table 2. Only the features that passed the significance test ($p < .01$) are listed in the tables.

We found a total of 16 significant features related to the pleasure and arousal of line expressions, and the features were divided into three levels of significance. The distribution of the feature classes is shown in Figure 7, which contains seven static features and nine motion features. Within all motion features, there are a further eight frequency domain features and a time domain feature. In terms of various original functions, there are two pressure features and two altitude features. The next section provides further details on a few of the key aspects.

### 6.2 How do hand-drawn lines express pleasure and arousal?

According to Tables 1 and 2, we can analyze the features of hand-drawn lines and their emotional expression.

Some of these features demonstrate how certain static features of lines are related to emotional metaphors.

S3 represents the y-coordinate value at the starting point of the line. It significantly negatively ($B = -0.003$, $p < .0001$) affects the pleasure of the hand-drawn line, indicating that the lower the starting point of the line, the higher the pleasure. Through observations of the samples and combining with the previous views, we believe that S3 may not affect the emotion of the line independently, but has a dependency relation with S2. The two can jointly explain the influence of the line trend on the line emotion. S2 is the direction of the image, which is related to three types of features: "tilt of the image, end coordinate of the image, average of velocity in the y direction". In feature set reduction, we used distance correlation to combine a number of features with correlation and select just one as the representative. This allowed us to trace its associated features and provide a better explanation. S2 represents the inclination of the image and significantly positively ($B = 0.040$, $p < .0001$) affects the pleasure of hand-drawn lines, indicating that the greater the inclination of lines, the higher the pleasure. This inclination means the angle between the main direction of the line and the horizontal direction. Although words of high pleasure, such as satisfaction, awe, adoration, and admiration, and negative emotions such as disgust and sadness, are represented by lines with a greater inclination, the

Table 1: Multiple Ordered Logistic Regression Results Based on Pleasure Dimension

| Feature Name | Brief Description | Sig. | Positive or negative correlation | Category | Motion component | Time or frequency domain | Order |
|---|---|---|---|---|---|---|---|
| S1(***) | Average of $y(t)$ | 0.000 | Positive | Static | - | - | - |
| S2(***) | Inclination of the image: the angle between the main direction of the line and the horizontal direction | 0.000 | Positive | Static | - | - | - |
| S3(***) | Start coordinate of the image: the y-coordinate value at the starting point of the line | 0.000 | Negative | Static | - | - | - |
| S4(*) | Zero-crossing rate of velocity in the x direction | 0.006 | Positive | Static | - | - | - |
| M1(**) | Median of the amplitude spectrum of $y(t)$: the median magnitude of the amplitude spectrum amplitude of the $y(t)$ transformed to the frequency domain | 0.001 | Negative | Motion | $y(t)$ | Frequency domain | 0 |
| M2(*) | Gravity frequency of the power density spectrum of the parametric estimation method for $y(t)$: the frequency of the centre of gravity of the power density spectrum obtained by transforming the $y(t)$ into the frequency domain with the parametric estimation method | 0.003 | Positive | Motion | $y(t)$ | Frequency domain | 0 |
| M3(*) | Mean square of the $p(t)$ energy spectrum : the mean square of the energy spectrum of the $p(t)$ transformed to the frequency domain | 0.002 | Negative | Motion | $p(t)$ | Frequency domain | 0 |
| M4(*) | Spurious-free dynamic range of the power density spectrum of the periodogram method for jerk in the x direction: unilateral SFDR of the power density map obtained using the periodogram method for the jerk function of $x(t)$ | 0.007 | Positive | Motion | $x(t)$ | Frequency domain | 3 |
| S5(**) | Average curvature of the image | 0.000 | Negative | Static | - | - | - |
| M5(**) | Occupied bandwidth of the power density spectrum of the parametric estimation method for spasm of pressure | 0.001 | Positive | Motion | $p(t)$ | Frequency domain | 4 |
| M6(*) | Mean square of the energy spectrum of velocity of altitude: the mean square of the energy spectrum obtained by converting the velocity function of altitude to the frequency domain | 0.002 | Positive | Motion | $al(t)$ | Frequency domain | 1 |

$*p<0.01, ** p <0.001, *** p <0.0001$

direction of the negative emotional line points to the lower right. Combining the relationship represented by S3 with that represented by S2, we can deduce that the lower the starting point of the line, the greater the gradient, and the higher the pleasure. This coincides with the phenomenon we have just mentioned. It is also an important conclusion that has been confirmed in the field of line emotion metaphors, which is that the greater the upper-right trend of the line, the greater the pleasure [40]. Important viewpoints in this field are extracted as salient features in our model, which validates the model's efficacy.

S5 represents the average curvature of the image and is correlated with two types of features: "the curvature of the image, the standard deviation of the curvature of the image". Its value is negatively correlated with the overall smoothness of the lines. S5 significantly negatively ($B$ =-2.122, $p$ <.001) affects the pleasure of the hand-drawn lines, indicating that the smoother the lines, the higher the pleasure. This finding coincides with the effect of human preference for curvature, which has been verified in academia multiple times [1,

3, 5, 7, 38]. Even with mathematical modelling and extensive feature extraction, we still find agreement with the earlier observation by Lundholm [33] that people tend to express pleasurable emotions with lines dominated by rounded corners. This once again validates the reliability of our model and further confirms that people tend to express high pleasure emotions with rounded lines.

S4 represents zero-crossing rate of velocity in x direction. It is correlated with four types of features: " zero-crossing rate of acceleration in the x direction, zero-crossing frequency of velocity in the x direction, the median frequency of the power density spectrum of periodogram method of velocity in the x direction, root mean square frequency of the energy spectrum of acceleration in the x direction, and root mean square frequency of the power spectrum of acceleration in the x direction ". It represents the number of velocity reversals in the horizontal direction. S4 significantly and positively ($B$ =8.859, $p$ <.01) affects the pleasure of hand-drawn lines, meaning that more velocity reversals in the x direction when drawing lines results in higher pleasure. Although this feature was calculated based

Table 2: Multiple Ordered Logistic Regression Results Based on Arousal Dimension

| Feature Name | Brief Description | Sig. | Positive or negative correlation | Category | Motion component | Time or frequency domain | Order |
|---|---|---|---|---|---|---|---|
| S1(***) | Average of $y(t)$ | 0.000 | Positive | Static | - | - | - |
| S3(**) | Start coordinate of the image :the y-coordinate value at the starting point of the line | 0.000 | Negative | Static | - | - | - |
| M7(***) | Frequency standard deviation of the power density spectrum of the parametric estimation method for the velocity in the y direction: the frequency standard deviation of the power density spectrum obtained by converting the velocity function corresponding to $y(t)$ to the frequency domain using the parametric estimation method | 0.000 | Negative | Motion | $y(t)$ | Frequency domain | 1 |
| S6(*) | Rectangularity of the image: the degree to which the object fills its outer rectangle | 0.002 | Negative | Static | - | - | - |
| M8(*) | Frequency standard deviation of the power density spectrum of the parametric estimation method for the velocity in the x direction: the standard deviation of the frequency of the power density spectrum obtained by converting the velocity function corresponding to $x(t)$ to the frequency domain using the parameter estimation method | 0.010 | Positive | Motion | $x(t)$ | Frequency domain | 1 |
| S7(**) | Skewness of $y(t)$: the symmetry of this signal | 0.001 | Positive | Static | - | - | - |
| M9(*) | Standard deviation of $al(t)$ | 0.010 | Positive | Motion | $al(t)$ | Time domain | 0 |

*$p<0.01$, ** $p<0.001$, *** $p<0.0001$

on the velocity function, observations of the original data suggest that it might still reflect an emotional metaphor for certain static features of lines. As we found that the subjects often used wavy lines resembling telephone lines to express pleasure, resulting in more velocity reversals in the x direction. Whereas such lines were rarely seen with low pleasure. This finding is in part still a variation of the a priori conclusion, that is, lines dominated by rounded corners express high pleasurable emotions, and lines dominated by sharp corners express low pleasurable emotions [33].

S1 is the average of $y(t)$ and is correlated with 17 types of features, typically including the "median of $y(t)$" and "mean square of $y(t)$". It significantly and positively affects the pleasure ($B =0.008$, $p <.0001$) and arousal ($B =0.007$, $p <.0001$) of hand-drawn lines. Based on our observations and samples, we suggest that this may represent an unmentioned idea, namely that the overall vertical position of highly pleasurable and aroused hand-drawn lines is closer to the top of the picture.

S7 is the skewness of $y(t)$, representing the symmetry of this signal. It significantly and positively ($B =0.335$, $p <.001$) affects the arousal of hand-drawn lines, implying that lines with lower arousal are likely to have more symmetry along the horizontal direction.

S6 is the rectangularity of the image, which represents the degree to which the object fills its outer rectangle. It significantly and negatively ($B =-2.179$, $p <.01$) affects the arousal of hand-drawn lines, indicating that lines with higher rectangularity had lower arousal. Based on our observations of the sample, we believe this may be because subjects tended to express the least arousing emotion with a horizontal line whose rectangularity is close to 1. The lines with higher arousal showed a more oscillatory pattern, with correspondingly lower rectangularity.

We next report the emotional metaphorical relationships for the motion features of the lines.

M9 is the standard deviation of the altitude and correlates with two types of features: "variance of altitude" and "shape factor of altitude". It significantly and positively ($B =17.027$, $p <.01$) affects the arousal of hand-drawn lines, meaning that the greater the disper-

sion of altitude when drawing a line, the higher the arousal. This suggests that human hand movements are more stable when drawing lines with lower arousal. M6 is the mean square of the energy spectrum obtained by converting the velocity function of altitude to the frequency domain. It correlates with 27 types of features, typically including "the variance of the energy spectrum of velocity of altitude" and "the median of the amplitude spectrum of acceleration of altitude". It significantly and positively ($B =0.000$, $p <.01$) affects the pleasure of the hand-drawn lines, meaning that the greater the vibrational energy of the velocity function of altitude change, the greater the pleasure. We suggest that this may represent, in part, that the greater the oscillation of altitude, the greater the pleasure. This is the first time that a relationship between altitude and line emotional metaphors has been found. The presence of altitude-related features in the significant features of both dimensional models implies that there is indeed a close link between altitude and the emotional expression of hand-drawn lines.

M3 is the mean square of the energy spectrum of the pressure function transformed to the frequency domain and correlates with the "variance of the energy spectrum of p(t)". It represents the mean square of the energy spectrum of $p(t)$ transformed to the frequency domain and significantly negatively ($B =-0.000$, $p <.01$) affects the pleasure of the hand-drawn lines. The smaller the mean square value of the energy spectrum of $p(t)$, the higher the degree of pleasure. The mean square value of the frequency domain amplitude responds to the magnitude of the vibrational energy. We believe that this relationship represents, to some extent, a negative correlation between the overall pressure of the subject's handwriting and pleasure. This is consistent with Lundholm's conjecture [33] about the relationship between line and pressure.M5 is the occupied bandwidth of the power density spectrum obtained by the parametric estimation method for the spasticity function of pressure. It significantly positively ($B =0.054$, $p <.001$) affects the pleasure of hand-drawn lines. This is the first time that a strong association between pressure data and line pleasure has been verified. Additionally, the linearity of the relationship has been established.

M4 is the spurious free dynamic range (SFDR) of the power density spectrum obtained of the periodogram method for jerk in the x direction. It represents the unilateral SFDR of the power density map as a function of the jerk in the x direction. The larger the SFDR, the lower the signal floor noise, which means the corresponding time domain signal is smoother. We assume that this represents, to some extent, the smoother the curve of the sharpness function in the x direction when drawing the line, the higher the pleasure level. This may indicate that the more stable the shift in hand acceleration in the horizontal direction, the higher the degree of pleasure. This is the first time that a relationship between jerk and pleasure has been identified.

Similar to M3, M4, M5, and M6 above, there are many other statistical features derived from motion data in the frequency domain that are more difficult to interpret directly as a specific sensation or behavior, such as M1, M2, M7, and M8.

M1 is the median of the amplitude spectrum of $y(t)$ and is correlated with 24 types of features, typically "the minimum of the amplitude spectrum of $y(t)$, the frequency of the gravity of the amplitude spectrum of $y(t)$". It represents the median magnitude of the amplitude spectrum amplitude of the $y(t)$ transformed to the frequency domain. And it significantly negatively ($B$ =-0.003, $p$ <.001) affects the pleasure of the hand-drawn lines.

M2 is the frequency of the gravity of the power density spectrum of the parametric estimation method for $y(t)$, and is correlated with three types of features, "the mean frequency of the power density spectrum of the parametric estimation method for $y(t)$, the median frequency of the power density spectrum of the parametric estimation method for $y(t)$ , and the occupied bandwidth of the power density spectrum of the parametric estimation method for $y(t)$". It represents the frequency of the centre of gravity of the power density spectrum obtained by transforming the $y(t)$ into the frequency domain with the parametric estimation method and symbolises the frequency of the signal components with larger components in the spectrum. It significantly positively ($B$ =7.816, $p$ <.01) affects the pleasure of the hand-drawn lines.

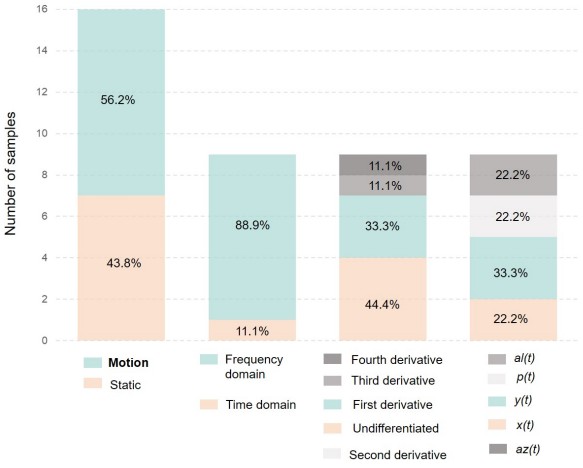

Figure 7: Features distribution

M7 is the frequency standard deviation of the power density spectrum of the parametric estimation method for the velocity in the y direction, which is related to the "frequency variance of the power density spectrum of the parametric estimation method for the velocity in the y direction". It represents the frequency standard deviation of the power density spectrum obtained by converting the velocity function corresponding to $y(t)$ to the frequency domain using the parametric estimation method, symbolising the dispersion

of the power spectrum energy distribution. It significantly negatively ($B$ =-0.830, $p$ <.0001) affects the arousal of the hand-drawn lines.

M8 is the frequency standard deviation of the power density spectrum of the parametric estimation method for the velocity in the x direction. And it is correlated with six types of features, typically "frequency variance of the power density spectrum of the parametric estimation method for the velocity in the x direction, occupied bandwidth of the power density spectrum of the parametric estimation method for the velocity in the x direction". It represents the standard deviation of the frequency of the power density spectrum obtained by converting the velocity function corresponding to x to the frequency domain using the parameter estimation method. It significantly positively ($B$ =0.517, $p$ <.01) affects the arousal of the hand-drawn lines.

Another area worth researching is the identification of the emotional metaphorical connexion between these potential features and intuitive sentiments. The discovery of the emotional metaphorical relationship between these features may not provide an intuitive answer to guide designers and artists in their work. But these findings have important implications in line with emotional computing or image emotional computing or its possible transfer. As an initial exploration of the frequency domain features of hand-drawn lines, their potential as important features to guide research related to line emotion computing. The high proportion of frequency domain features illustrated in Figure 7 implies that frequency domain features are a non-negligible part of line emotional computing.

In summary, as a first exploration of the link between line motion features and emotion, some interesting findings were obtained. Some static features were found to be consistent with previous studies [3, 33, 40], confirming the validity of our model. In terms of motion features, we verified for the first time a significant association between pressure and the pleasure of the lines. And we proposed a linear relationship between altitude during drawing and the pleasure and arousal of the lines. This is the first time that altitude was found to be significantly associated with the emotional expression of the lines. In addition, many frequency domain features that are difficult to interpret as intuitive feelings were found to be salient. That implies that we may have found some extremely potential relationships between the features of the line and emotional expression. The high proportion of motion and frequency features in the set of significant features (Figure 7) suggests that it is necessary to use them in lines' emotional metaphor studies and emotional computing.

## 7 CONCLUSIONS AND FUTURE PERSPECTIVES

This paper explored the significant factors that influence the emotional expression of hand-drawn lines, specifically by expanding on the motion data. The abundant static, time-domain, and frequency-domain features of hand-drawn lines were calculated, and linear relationships were found between arousal, pleasure, and key features that significantly affect line emotion through multiple ordered logistic regression modelling. The discovered significant features include static features, motion features, and frequency-domain features. The findings based on static features overlap with the a priori findings, validating our model and data. The findings based on motion features reveal for the first time the mapping relationship between pressure, altitude, and hand-drawn lines. The frequency-domain features may represent the significant influence of certain potential features on the presence of line emotion.

The mapping relationships between features and pleasure and arousal found in this paper can serve as a guide for artists, particularly multimedia artists based on interactive surfaces. It enriches the research results in line cognitive emotion, especially with regard to the emotional metaphors of motion features of lines. The dataset and important features obtained in this paper can be used later to build line emotion recognition models. Furthermore, their transfer to the computation of emotion in line, picture, digital painting, and

interactive art related to pen movement is an important area that can be extended in the future.

## ACKNOWLEDGMENTS

We thank all the participants for their participation and feedback. This work was supported by Postgraduate Research & Practice Innovation Program of Jiangsu Province, the 2022 Open Project of Jiangsu Key Laboratory of Media Design and Software Technology (Jiangnan University), and Humanities and Social Sciences Foundation of Ministry of Education of China (18YJC760123).

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
