# OpenReview forum: "Critical influence of implicit motion features on hand-drawn lines’ emotional expression"
_graphicsinterface.org/Graphics_Interface/2023/Conference — GI 2023_

### Official Review · Reviewer_YGjy · 2023-01-13
**Interesting work; subpar writing**

**Rating:** 7
**Confidence:** 3

**Review:**

The paper presents a study of how humans express emotions via drawing lines. In their main experiment, subjects are given emotional words and are asked to drawn a line on a tablet, expressing that emotion. The system captured the physical trajectories and all the extra properties of the strokes. The authors found correlations between the 'pleasure' or 'arousal' dimensions of the emotion with some of the physical properties of strokes.

In general, the study is done well, and certainly provides an interesting insight into artistic practice. The only critique on the study design I have is that I somehow have not found statistics how many participants are professional or experienced artists -- I would expect that to influence the way they express emotions.

My only complaint is mostly the writing: the introduction in particular is written in a bit of a vague and cryptic way. For example, the 'motion' of a stroke is neither properly defined nor illustrated, even though the authors rely on that terminology.
Furthermore, the intro often uses wording that is, at best, confusing:
	- "cognitive emotion" -- what is that?
	- "implicit motion features" -- never defined
	- "passive emotion induction" -- ?

Some phrasing or entire sentences also should be rewritten:
	- "For example, Degas, … etc." -- this is not a complete sentence
	- "it makes us feel that they will be more human" -- both grammar and phrasing seem incorrect
	- "it cannot cover as many emotion categories as possible" -- awkward phrasing

Again, it's a good work in general, but the text, in particular, the intro needs to be heavily edited.

---

### Official Review · Reviewer_tKCV · 2023-01-14
**A thorough analysis but the paper is unclear on several key points.**

**Rating:** 6
**Confidence:** 4

**Review:**


This paper analyzes how people express different emotions using lines. The authors collect motion data from participants using a digital pen tablet system and then model the relationship between line characteristics and the expressed emotion using multinomial logistic regression. Unlike previous works, the authors use both static and motion features in their analysis.

The authors use an interesting set of features in their analysis. Their findings may have parallels with other areas of affective computing, such as human gestures.

The paper is not immediately clear that they study how people express emotions through lines, rather than how people perceive emotions in lines. It wasn't until section 3.1.3 that I realized this distinction. Regardless, I would be very interested
to know whether people perceive the emotions that the participants expressed.

From the results and discussion, I'm also not clear on how these results can be used. Do the motion features tell us anything that we can't already infer from static features?

In section 3, what specific motion features have been previously studied, and which do you add to this study?

In Figure 5, did 7 correspond to low arousal and 1 to high
arousal? Were the measurement axes flipped for your study?

In section 5.2, how were velocities calculated (backward differencing? central differencing? How many points were used)? What types of general statistics features did you use? What were the 4 independent time domain features? Can you specifically say which features you used from reference [10]? The paper would be improved if the features were explicitly defined. I realize there are a large number of initial features, but many could be described in terms of a single curve, such as x(t). Perhaps this could be in a supplemental document.

In section 5.3, how did you choose which of the co-linear features to use in the regression model?

In section 6.2, could you give some intuition as to why some features are correlated? Do they measure similar aspects? Because the majority of features are not explicitly defined, many of the correlated features do not make sense. For example, what is the "excess rate of the x(t) 1st order derivative correlated with the root mean square frequency of the energy spectrum of the x(t) second derivative"? What is the "shape factor of altitude"?

The paper may read more easily if you used the terms "x-velocity" and "x-acceleration", rather than x(t) 1st and 2nd derivatives.

I found various typos in the text, such as, "needs satisfies", "that makes us can retrace", and "we assume represents".

---

### Official Review · Reviewer_GJuX · 2023-01-14
**A well written and interesting paper**

**Rating:** 8
**Confidence:** 3

**Review:**

So, I found the paper really interesting - possibly the biggest problem with it is that it covered 3 different topic areas (analysis, art/drawing, and psychology) and I am not sure many people cross over all disciplines in a way that can provide a full review.

Overall the paper is written very competently, it is laid out and discussed in a very clear manner and even though drawing/emotional response are not topics I am familiar with in themselves, I was able to understand clearly. I will say that I have no emotional response to the lines produced nor am I sure I would produce the same lines given the task; but that isn't really my thing anyway. There are a few grammatical mistakes (not spelling) which a quick review will clear up (an example, "The subject was seating in a chair" should either be "The subject was seated in the chair" or "The subject was sitting in the chair") ... this didn't really make the paper less readable, but should be cleared up.

The number of participants in the study was impressive, but needed to achieve the correlation confidence required. However, it was not clear if the participants had prior knowledge of the way that lines express specific emotion. The authors discuss at length the way this has been studied (in various forms) and even discuss famous painters (which was also interesting), yet it was not clear if the participants had knowledge of this and thus influences/biased the results  ... a grouping might have been useful in this case. In addition, it was mentioned that around 48% were female, and 52% were male, but there was no further discussion on how this influenced the results - and I think this would have helped to understand the results better (if it made any difference) or a statement indicating it made no difference.

I would select a different colour for the graph in Figures 3 and 7, as when printed in B&W it didn't come out clearly and only upon reviewing the PDF did I notice it more clearly -  a small detail, but important for printed documents and e-Readers.

Overall though I cannot really comment further, I think it's a well written paper, clear results, and nicely presented. Apart from the grammatical errors, I would say it needs little attention before submission.

---

### Meta-Review · Area_Chair_h471 · 2023-01-15

**Recommendation:** 7
**Confidence:** 4

**Metareview:**

The reviewers agree that this paper presents an interesting and thorough analysis of how people express emotions using lines, based on motion features.

Before publication, the authors should integrate the feedback and requests for greater detail so that the paper is both easier to read and reproducible by another person. In particular,

- Be clearer that the authors study emotional expression via lines.
- More details about how features are computed.
- Better definitions and introductions of terms.
- Fix typos throughout the text.
- Check axes and colors of all figures, especially Figures 3, 5, and 7